# Genome Mining Discovery of a New Benzazepine Alkaloid Pseudofisnin A from the Marine Fungus *Neosartorya pseudofischeri* F27-1

**DOI:** 10.3390/antibiotics11101444

**Published:** 2022-10-20

**Authors:** Xiao-Xin Xue, Lin Chen, Man-Cheng Tang

**Affiliations:** 1State Key Laboratory of Microbial Metabolism, School of Life Sciences and Biotechnology, Shanghai Jiao Tong University, Shanghai 200240, China; 2Zhangjiang Institute for Advanced Study, Shanghai Jiao Tong University, Shanghai 200240, China

**Keywords:** 1-benzazepine, l-kynurenine, genome mining, biosynthesis, iterative methyltransferase

## Abstract

l-Kynurenine (Kyn) is an intermediate in the kynurenine pathway and is also found to be a building block or biosynthetic precursor to bioactive natural products. Recent studies revealed that l-Kyn can be incorporated via nonribosomal peptide synthetase (NRPS) biosynthetic routes to generate 1-benzazepine-containing compounds, while 1-benzazepine is a pharmaceutically important scaffold that is rarely found in natural products. Using a core biosynthetic enzyme-guided genome-mining approach, we discovered a biosynthetic gene cluster from *Neosartorya pseudofischeri* and identified that it encodes for the biosynthesis of pseudofisnins, novel 1-benzazepine-containing compounds. The biosynthetic pathway of pseudofisnins was elucidated through in vivo and in vitro experiments. The methyltransferase PseC from the pathway was biochemically characterized to be an iterative methyltransferase that catalyzes off-NRPS line di-methylation on an amine group.

## 1. Introduction

l-Kynurenine (Kyn) (Figure 1A) is a metabolite of the l-tryptophan (Trp) metabolism and is known as the biosynthetic precursor to synthesize nicotinamide adenine dinucleotide in the well-known kynurenine pathway [1]. The conversion of l-Trp to *N*-formyl-l-Kyn, which is rapidly hydrolyzed to l-Kyn, is catalyzed by indoleamine-2,3-dioxygenase (IDO) or tryptophan-2,3-dioxygenase (TDO) [2,3]. l-Kyn is present in milk proteins, lens crystallins, and human Cu^2+^/Zn^2+^ superoxide dismutase in vivo [1]. In addition, l-Kyn is a non-proteinogenic amino acid with roles in a number of biochemical signaling pathways [4,5]. However, l-Kyn is rarely found as a building block or biosynthetic precursor to bioactive natural products. In the past few decades, a limited number of examples have been reported. For example, l-Kyn is a building block of daptomycin, which is a lipodepsipeptide isolated from *Streptomyces roseoporus* that is used in the treatment of Gram-positive pathogen skin infections [6]. Recently, several research groups reported the genome mining discovery of novel natural products, such as nanangelenin A and aspcandine, from fungi with new chemical scaffolds derived from l-Kyn, which is incorporated by nonribosomal peptide synthetase (NRPS) [7,8,9]. An interesting finding in these studies is that all the identified gene clusters contain genes encoding for NRPS and IDO. Considering the accumulation of fungal genome sequence data, the discovery of novel l-Kyn-derived bioactive compounds could be achieved through the mining of the available fungal genomes. 

Benzazepines are a class of nitrogenous heterocyclic compounds with diverse chemical structures and broad bioactivities, exemplified by the heart-rate-lowering agent ivabradine and the angiotensin-converting enzyme inhibitor, benazepril [10,11]. Natural products containing a benzazepine structural unit are found to be widespread in nature, especially the 2- and 3-benzazepines, such as communesin [12] and rhoeadine [13]. However, 1-benzazepine-containing natural products are relatively rare. In recent studies, novel 1-benzazepine-containing compounds have been discovered from different fungal species via genome mining [7,8,9]. Biosynthetic studies have revealed that the 1-benzazepine unit within these compounds was directly derived from l-Kyn by NRPS pathways (Figure 1B). Since there are many cryptic NRPS biosynthetic gene clusters that exist in the fungal genomes, we reasoned that other novel 1-benzazepine-containing natural products could be discovered through genome mining. 

Here, we reported the discovery of pseudofisnins, 1-benzazepine-containing compounds from the marine fungus *Neosartorya pseudofischeri* F27-1, via a core biosynthetic enzyme-guided genome mining approach. A biosynthetic pathway to generate these compounds was established by both in vivo and in vitro experiments. In addition, a methyltransferase PseC from the pathway was characterized to be an iterative methyltransferase catalyzing off-NRPS line methylation modification. 

## 2. Results and Discussion

### 2.1. Identification of the Pse Cluster through Core Biosynthetic Enzyme-Guided Genome Mining 

As reported in the biosynthesis of nanangelenins, the core biosynthetic enzymes involved in the nanangelenin scaffold assembly are the NRPS NanA and the IDO NanC [7]. A number of cryptic biosynthetic gene clusters (BGCs) can also be found in the sequenced fungal genomes, using NanA and NanC as the search query [7]. Therefore, we reasoned that the combination of NanA and NanC homolog pairs with different tailoring enzymes could lead to the generation of novel compounds belonging to the benzazepine alkaloid family. 

Using the core biosynthetic enzymes, NanA and NanC, as the search query, we identified a compact BGC *pse* cluster from the genome of the marine fungus, *N. pseudofischeri* F27-1 (Figure 2A). Apart from the genes encoding NanA and NanC homologs, *pseA* and *pseB*, respectively, the *pse* cluster also encodes a methyltransferase (*pseC*), a flavine-dependent monooxygenase (*pseD*), a hypothetical protein (*pseE*), a deaminase (*pseF*), and a UbiH type hydroxylase (*pseG*). Since the other genes (*pseC*-*G*) of the *pse* cluster are different from the tailoring genes in the nanangelenin biosynthetic pathway, we hypothesized that the *pse* cluster might encode for novel benzazepine-containing natural products. 

### 2.2. Heterologous Expression of the Pse Cluster and Characterization of the Products 

To explore the natural product encoded by the *pse* cluster, all seven putative biosynthetic genes (*pseA*-*G*) were introduced into an engineered *Aspergillus nidulans* expression host on three episomal vectors (Appendix A) [14]. Compared to the negative control, two new metabolites, **1** (with a molecular weight of 337) and **2** (with a molecular weight of 323), were identified from the extract of *A. nidulans* expressing *pseA*-*G* (Figure 2B, trace (ii), and Appendix A). For the purposes of structural elucidation, these two compounds were isolated and purified. Based on 1D and 2D Nuclear Magnetic Resonance (NMR) spectroscopy data (Appendix A), **1** (named pseudofisnin A) and **2** (named pseudofisnin B) were characterized to be 1-benzazepine-containing compounds (the structures of which are shown in Figure 2C). The only difference between **1** and **2** is that the amine group attached to the benzene ring is di-methylated in **1** but mono-methylated in **2**. 

The characterization of **1** and **2** validated our hypothesis that the *pse* cluster encodes the enzymes for the biosynthesis of natural products belonging to the benzazepine alkaloid family. In the previously reported studies of nanangelelins, nanangelenin B (**3**, the structure shown in Figure 2C) was identified to be the product of the NRPS assembly line and the biosynthetic intermediate to other nanangelelin compounds [7]. As **1** and **2** are structural analogs to **3**, we reasoned that **3** could also be the biosynthetic precursor to **1** and **2**. Combined with the bioinformatics analysis results, we proposed that the biosynthesis of pseudofisnins would involve the IDO encoding gene, *pseB*, the NRPS encoding gene, *pseA*, and one or two methyltransferase encoding genes, such as *pseC*. 

### 2.3. Functional Identification of the Genes Involved in the Biosynthesis of Pseudofisnins

To test our hypothesis for the biosynthesis of pseudofisnins, different combinations of other genes from the *pse* cluster were co-expressed together with *pseA* and *pseB* in *A. nidulans*. Interestingly, when the putative methyltransferase PseC was co-expressed with PseA and PseB, the production of **1** and **2** was detected (Figure 2B, trace (iii)). When *pseC* was removed from the co-expression strain, the production of **1** and **2** was abolished (Figure 2B, trace (iv)). Instead, another new metabolite (**3**) with an MW of 309 emerged, which is identical to the MW of nanangelenin B. Structural elucidation of this new metabolite, based on NMR data (Appendix A), confirmed it to be nanangelenin B.

These results confirmed that only three genes, encoding for the NRPS PseA, the IDO PseB, and the methyltransferase PseC, are required for the biosynthesis of pseudofisnins, which is consistent with our hypothesis. Similar to the biosynthesis of nanangelenins, PseA and PseB are responsible for the biosynthesis of nanangelenin B. In addition, the results from the heterologous expression studies suggested that the methyltransferase PseC might catalyze the methylation of **3** to afford **1** and **2**, acting as an iterative methyltransferase. 

### 2.4. In Vitro Characterization of PseC as an Iterative Methyltransferase

To further validate the function of PseC, the intron-free *pseC* gene was cloned from the extracted cDNA of *A. nidulans,* expressing the *pse* cluster and being overexpressed in *Escherichia coli* BL21 (DE3) as an *N*-terminal His_6_-tagged fusion protein. In the presence of PseC and the cofactor, *S*-adenosyl methionine (SAM), in PIPES (1,4-piperazinediethanesulfonic acid) buffer (pH 7.4), the conversion of **3** to **1** and **2** was clearly detected after 1 h of incubation (Figure 3, trace (iv)). To further investigate, **2** was tested as the substrate under the same reaction conditions. As shown in Figure 3, **2** could be totally converted into **1** by PseC (Figure 3, trace (ii)). These results confirmed that PseC is an iterative methyltransferase catalyzing di-methylation on the amine group of **3** to **1** by a stepwise mechanism. 

To explore the catalytic efficiency of PseC in the first methylation step and the second methylation step, kinetic studies of PseC on different substrates were carried out under optimized reaction conditions. When **3** was tested as the substrate, to minimize the formation of **1** and accumulate **2** as the major product, the enzymatic reaction mixtures were only incubated for 10 min at 30 °C. As shown in Appendix A, the kinetic measurements showed PseC to have a *K_M_* of 72 ± 16 µM and *k_cat_* of 9.8 ± 0.8 min^−1^ toward **3** for the first methylation step, and a *K_M_* of 14 ± 5 µM and *k_cat_* of 0.29 ± 0.04 min^−1^ toward **2** for the second methylation step. The results indicated that PseC has a stronger binding affinity toward **2** over **3**, which could suggest that PseC will efficiently catalyze the second methylation step to complete the biosynthesis, to form **1**.

### 2.5. The Proposed Biosynthetic Pathway of ***1***

By combining the results from both in vivo and in vitro experiments, a plausible biosynthetic pathway to **1** was proposed. As shown in Figure 4, the IDO homolog PseB could catalyze the ring-opening oxidation of l-Trp to form *N*-formyl-l-Kyn, which is then hydrolyzed to form l-Kyn, either spontaneously or via the unknown endogenous kynurenine formamidase. Then, the NRPS PseA incorporates one molecule of anthranilic acid and one molecule of l-Kyn to assemble the dipeptide scaffold, which is further released by the C_T_ domain through regioselective lactamization to form **3,** with a benzazepine core structure. Finally, the iterative methyltransferase PseC catalyzes the stepwise methylation of **3**, to complete the biosynthesis of **1**. 

## 3. Conclusions

In summary, by combining genome mining and heterologous expression, a biosynthetic gene cluster, including IDO and NRPS encoding genes, was mined from the genome of the marine fungus *N. pseudofischeri* and was characterized as being responsible for the biosynthesis of the novel benzazepine alkaloid, pseudofisnin A. The biosynthetic pathway of pseudofisnin A was proposed and established based on in vivo and in vitro studies. An iterative methyltransferase was biochemically characterized to catalyze the off-line post-di-methylation on the amine group of nanangelenin B, to complete the biosynthesis of pseudofisnin A. Our results provide a successful example of the discovery of novel natural products through core biosynthetic enzyme-guided genome mining. 

## 4. Materials and Methods

### 4.1. Strains and Culture Conditions 

The *Neosartorya pseudofischeri* strain F27-1 was grown at 28 °C over 4 days on PDA medium (Potato Dextrose Water, 2% agar) for genome extraction. The *Saccharomyces cerevisiae* strain BJ5464-NpgA (*MATα ura3-52 his3-Δ200 leu2-Δ1 trp1 pep4::HIS3 prb1 Δ1.6R can1 GAL*) was used for in vivo homologous recombination and was cultured in YPD (yeast extract 1%, peptone 2%, glucose 2%) media or SDCt (uracil dropout) medium at 28 °C. *Aspergillus nidulans* A1145 *ΔEMΔST* [15] was used as the host for heterologous expression, cultured at 37 °C on CD medium (1% glucose, 50 mL/L of 20× nitrate salts, 1 mL/L of trace elements, pH 6.5) for 3 days for sporulation, or fermented at 28 °C in CD-ST (starch 2%, tryptone 2%, 50 mL/L of 20× nitrate salts, 1 mL/L of trace elements, pH 6.5). The 20× nitrate salts solution was prepared by dissolving 120 g NaNO_3_, 10.4 g KCl, 10.4 g MgSO_4_·7H_2_O, and 30.4 g KH_2_PO_4_ in 1 L double-distilled water. The trace element solution (100 mL) consisted of 2.20 g of ZnSO_4_·7H_2_O, 1.10 g of H_3_BO_3_, 0.50 g of MnCl_2_·4H_2_O, 0.16 g of FeSO_4_·H_2_O, 0.16 g of CoCl_2_·5H_2_O, 0.16 g of CuSO_4_·5H_2_O, and 0.11 g of (NH_4_)_6_Mo_7_O_24_·4H_2_O, and the pH was adjusted to 6.5. All *Escherichia coli* were cultured in LB medium at 37 °C. *E. coli* BL21 (DE3) (Tsingke, Beijing, China) was used for protein expression. 

### 4.2. General DNA Manipulation Techniques

The genomic DNA of *N. pseudofischeri* F27-1 was extracted using the CTAB method [16]. *E. coli* HB101 and DH5α were used for cloning, following the standard recombinant DNA techniques. DNA restriction enzymes were used as recommended by the manufacturer (New England Biolabs, NEB, Ipswich, MA, USA). PCR reactions were performed using Phanta^®^ Max Super-Fidelity DNA Polymerase (Vazyme, Nanjing, China). The gene-specific primers are listed in Appendix A. The plasmids (Appendix A) that are used for heterologous expression in *A. nidulans* were constructed by yeast homologous recombination using the Frozen-EZ Yeast Transformation II Kit™ (Zymo Research, Irvine, CA, USA). The yeast plasmid extraction was performed with Zymoprep™ Yeast Plasmid Miniprep I (Zymo Inc., Irvine, CA, USA). 

### 4.3. Plasmids Construction

The vectors pYTP, pYTU, and pYTR, with auxotrophic markers for pyridoxine (*pyroA*), uracil (*pyrG*), and riboflavin (*riboB*), respectively, were used for insertion into target genes. All three plasmids were digested with *Pac*I and *Swa*I. The target genes were amplified from the genomic DNA of *N. pseudofischeri* F27-1, with their native terminators added to the ends, and the amplified fragments were flanked by homologous arms for recombination. The prepared fragments and vectors were co-transformed into *S. cerevisiae* BJ5464-NpgA yeast-competent cells to yield the plasmids for gene expression in *A. nidulans*, which were verified by sequencing.

To construct the protein expression plasmid of PseC for *E. coli* BL21 (DE3), the intron-free *pseC* gene was amplified from the cDNA of the *A. nidulans* strain expressing the *pse* gene cluster, using the primers listed in Appendix A. The expression vector pET28a was digested with *Nde*I and *Bam*H I. The digested vector and the target fragment with homologous arms were recombined using the ClonExpress^®^ II One-Step Cloning Kit (Vazyme). The resulting plasmid pXXX1-7 (Appendix A) was verified by enzyme digestion and sequencing.

### 4.4. Protoplast Preparation and Transformation of A. nidulans 

The spores of *A. nidulans* A1145 *ΔEMΔST* were inoculated into 50 mL of liquid CD, containing 10 mM uridine, 5 mM uracil, 0.5 µg/mL pyridoxine and 0.125 µg/mL riboflavin, and then germinated at 37 °C at 220 rpm for approximately 9 h. The mycelia were harvested by centrifugation at 5000 rpm for 5 min at 4 °C and washed with 15 mL osmotic buffer (1.2 mol/L MgSO_4_·7H_2_O, 10 mM sodium phosphate, pH 5.8). The mycelium was then transferred to 10 mL osmotic buffer containing 30 mg lysing enzymes and 20 mg Yatalase and was then shaken at 28 °C and 80 rpm for 14 h. After the protoplasts were observed under the microscope, the liquid was poured into a sterile 50 mL centrifuge tube and gently covered with an equal volume of trapping buffer (0.6 M sorbitol, 0.1 M Tris-HCl, pH 7.0). Centrifugation at 3750 rpm for 15 min at 4 °C layered the protoplasts at the interface of the two buffers. Then the protoplasts were washed and resuspended with STC buffer (1.2 M sorbitol, 10 mM CaCl_2_, 10 mM Tris-HCI, pH 7.5).

For transformation, recombinant plasmids were added to the protoplasts of *A. nidulans* and placed on ice for 30 min. First, 60% PEG buffer (60% PEG 4000, 0.56% CaCl_2_, 50 mM Tris, pH 7.5) was added to the mixture and then induced at room temperature for 1 h. The mixture was grown on the regeneration dropout solid medium (CD medium with 1.2 mM sorbitol and appropriate supplements) at 37 °C for about 2 days. The transformants were inoculated on fresh dropout solid medium at 37 °C for about 2 days.

### 4.5. Chemical Analysis and Compound Isolation and Characterization

In order to perform small-scale metabolite analysis in *A. nidulans*, different transformants were grown on liquid CD-ST media at 28 °C for 3–5 days, then extracted with ethyl acetate (EtOAc). UPLC–MS analysis was carried out for 15 min on a Shimadzu LC-30A system connected to a single quadrupole mass spectrometer MS2020 (ESI) (Shimadzu, Kyoto, Japan), using a C18 reverse-phase column (shim pack XR-ODS III, 2.0 mm × 75 mm, 1.6 μm) with a linear gradient of 5−95% MeCN-H_2_O and a flow rate of 0.2 mL/min.

For the isolation of nanangelenin B (**1**), pseudofisnin A (**2**), and pseudofisnin B (**3**), the corresponding transformants of *A. nidulans* strains were grown on 4 L of liquid CD media for 4 days at 28 °C and then extracted with EtOAc. After concentration, the crude extract was chromatographed on a silica gel column using EtOAc and *n*-hexane as eluent. The fractions containing the target compounds were combined and further purified by semi-preparative HPLC, using a Pntulips C18 reverse-phase column (C18, 5 μm, 250 × 10 mm) (Puningtech, Shanghai, China).

The 1D and 2D NMR spectra were recorded in CDCl_3_ or DMSO-*d_6_* using Bruker 600 MHz spectrometers and tetramethylsilane (TMS) as an internal standard. HR-ESIMS (high-resolution electrospray ionization mass) data were measured on an Acquity 2D-UPLC/Acquity UPC2/Xevo G2-XS QTOF (Waters, Milford, MA, USA).

### 4.6. Protein Expression and Purification of PseC

The correct plasmid was transformed into BL21 (DE3) to express the *N*-terminal His_6_-tagged fusion protein. The protein purification of PseC was performed as follows. The transformed strains were first cultured overnight in 5 mL LB containing 50 µg/mL kanamycin and were then transferred to 2 L LB medium containing antibiotics at a ratio of 1:100. The culture was shaken at 37 °C until the optical density at 600 nm (OD_600_) reached 0.6~0.8, and then 0.2 mM isopropylthio-β-D-galactoside (IPTG) was added for induction. After induction at 16 °C for 22 h, the cells were collected via centrifugation at 5000 rpm. The collected cells were resuspended in lysis buffer (20 mM Tris-HCl, 300 mM NaCl, 5 mM β-mercaptoethanol, 10 mM imidazole, 1 mM PMSF, 10% glycerol, pH 8.0) and sonicated on ice. After centrifugation (12,000 rpm, 30 min, 4 °C), the supernatant was collected and added to Ni-NTA agarose resin (Smart-Lifesciences, Changzhou, China). Then, the resin was loaded into a gravity flow column and eluted in a gradient with elution buffer (20 mM Tris-HCl, 300 mM NaCl, 10% glycerol, pH 8.0) containing progressively increasing concentrations of imidazole (20 mM to 300 mM). After analysis by 12.5% (*w*/*v*) acrylamide sodium dodecyl sulfate polyacrylamide gel electrophoresis (SDS-PAGE) (Appendix A), the protein fractions containing PseC were combined and concentrated using Amicon-Ultra Centrifugal Filters (Millipore, Burlington, MA, USA). The protein solution was concentrated to 2.5 mL and further desalted using a PD-10 column (GE Healthcare, Chicago, IL, USA) with elution buffer (50 mM HEPES, 100 mM NaCl, 10% glycerol, pH 7.5). The concentration of the protein was measured with a Bradford protein assay (Bio-Rad, Hercules, CA, USA). Finally, the proteins were flash-frozen in liquid nitrogen and stored at −80 °C.

### 4.7. Enzymatic Assays of PseC In Vitro

To verify the function of PseC in vitro, 10 μM PseC was incubated with 120 µM nanangelenin B (**1**) or 30 µM pseudofisnin A (**2**), with 2 mM SAM in 50 mM PIPES (pH 7.4) buffer. The reaction mixture was incubated at 30 °C for 1 h and then quenched with 2 times the volume of acetonitrile. Boiled PseC was used in the negative control assays. After centrifugation, the supernatant was analyzed via LC-MS, using the method mentioned above.

Kinetic studies of the protein were performed using the following conditions: 1 µM PseC was incubated with 4–480 µM nanangelenin B (**1**) or 2.5–115 µM pseudofisnin A (2), with 2 mM SAM in 50 mM PIPES (pH 7.4) buffer for 10 min at 30 °C. After quenching with acetonitrile, the reaction mixtures were analyzed by LC-MS to determine the product formation. Each data point represents a minimum of three replicate endpoint assays; kinetic constants were obtained by non-linear regression analysis using GraphPad Prism 9 (GraphPad Software, Inc., San Diego, CA, USA).

## Figures and Tables

**Figure 1 antibiotics-11-01444-f001:**
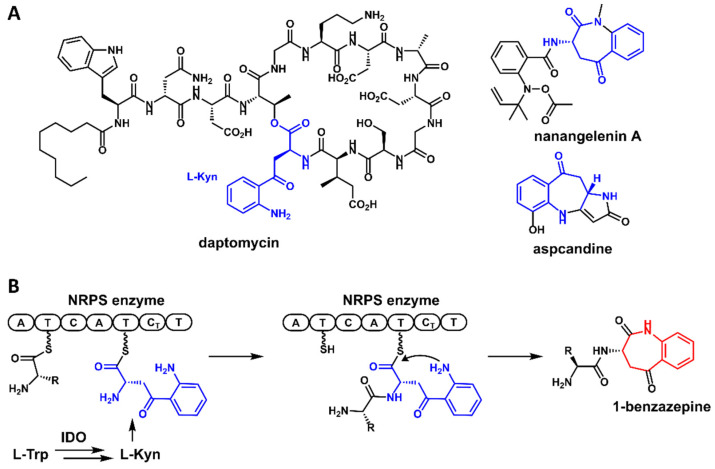
The structures of l-kynurenine (l-Kyn)-derived natural products (**A**) and the representative biosynthetic route to generating 1-benzazepine-containing compounds from l-Kyn (**B**).

**Figure 2 antibiotics-11-01444-f002:**
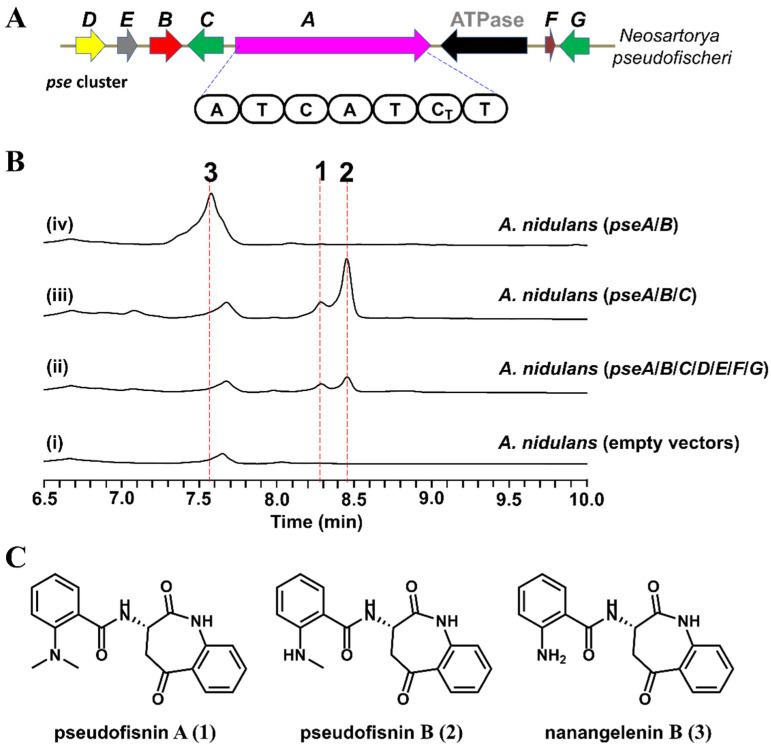
The *pse* cluster is responsible for the biosynthesis of pseudofisnins. (**A**) The *pse* cluster identified from *Neosartorya pseudofischeri*. *pseA* encodes NRPS, *pseB* encodes IDO, *pseC* encodes methyltransferase, *pseD* encodes flavine-dependent monooxygenase, *pseE* encodes a hypothetical protein, *pseF* encodes deaminase, and *pseG* encodes UbiH type hydroxylase. (**B**) Product profiles of *A. nidulans,* transformed with combinations of *pse* genes. The traces are HPLC with λ = 312 nm. (**C**) The structures of the characterized compounds.

**Figure 3 antibiotics-11-01444-f003:**
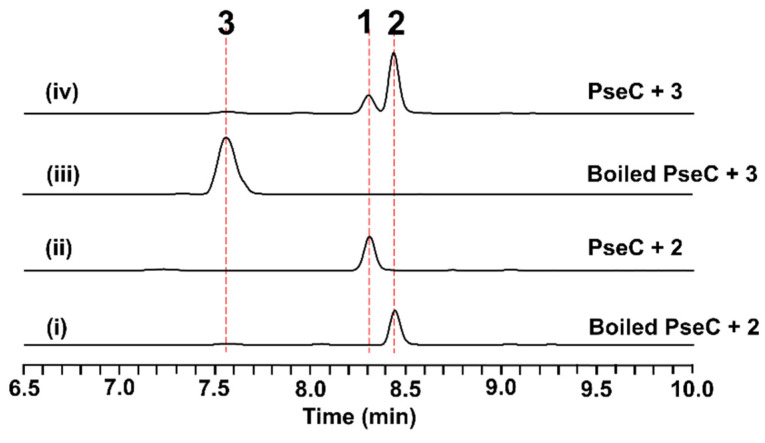
Biochemical characterization of the methyltransferase PseC in vitro. The traces are of HPLC with λ = 312 nm.

**Figure 4 antibiotics-11-01444-f004:**
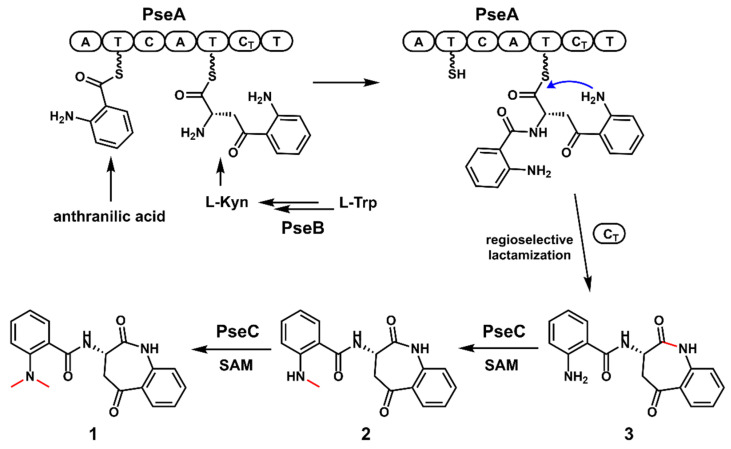
The proposed biosynthetic pathway of pseudofisnins.

## Data Availability

Data are available in publicly accessible repository and within the article or Appendix A.

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
