# Peer review of "Genome Mining Discovery of a New Benzazepine Alkaloid Pseudofisnin A from the Marine Fungus Neosartorya pseudofischeri F27-1"

_antibiotics, 2022, doi:10.3390/antibiotics11101444_

Round 1

Reviewer 1 Report

The manuscript antibiotics-1975547 entitled "Genome Mining Discovery of a New Benzazepine Alkaloid Pseudofisnin A from the Marine Fungus Neosartorya pseudofischeri F27-1" discovered a biosynthetic gene cluster responsible for the biosynthesis of a novel benzazepine alkaloid pseudofisnin A. The subject is really very interesting. The authors have well proposed the pseudophysnin A biosynthetic pathway identifying the genes involved and their functions. The manuscript is well constructed and well written. Methods and results are clearly explained, and the discussion is argued and balanced. 

Please see my minor revisions below.

L60: PseC genes? Enzymes?

L189: Change “isolated” to “extracted”. 

How did the authors assess DNA integrity and quantity?

Author Response

Many thanks for the reviewer's comments.

L60: PseC genes? Enzymes?

Response: PseC is an enzyme, the methyltransferase biochemically characterized in this study. The gene name is pseC.

L189: Change "isolated" to "extracted".

Response: Revised as suggested.

How did the authors assess DNA integrity and quantity?

Response: We used agarose gel electrophoresis to assess DNA quality or integrity. In addition, we used Nanodrop to assess DNA purity and quantity.  

Reviewer 2 Report

Xue et al. report the identification and characterization of a gene cluster in the organism Neosartorya pseudofischeri. They identify the enzymes that participate in the biosynthesis of three compounds that contain the 1-benzazepine group. They biochemically characterize a methyltransferase that participates in this pathway. This manuscript is very well structured and contains all required information. I have only very minor suggestions for changes.

Recommended changes:

1. Figure 1a shows the formula of aspcandine but this molecule is not mentioned in the text. Either add a discussion in the text or remove this figure panel to avoid confusing the reader.

Typos:

Line 263: ‘After analyzed’ should probably mean ‘After analysis’

L. 280: ‘After quenched’ should probably be ‘After quenching’

Author Response

Many thanks for the reviewer's comments.

1. Figure 1a shows the formula of aspcandine but this molecule is not mentioned in the text. Either add a discussion in the text or remove this figure panel to avoid confusing the reader.

Response: Many thanks. We have revised the maintext to mention aspcandine.

"Recently, several research groups reported the genome mining discovery of novel natural products, such as nanangelenin A and aspcandine, from fungi with new chemical scaffold derived from L-Kyn that is incorporated by nonribosomal peptide synthetase (NRPS) [7-9]."

2. Line 263: "After analyzed" should probably mean "After analysis".

Response: Thanks. Revised as suggested.

3. Line 280: "After quenched" should probably be "After quenching".

Response: Thanks. Revised as suggested.